# High Fidelity Video Prediction with Large Stochastic Recurrent Neural Networks

**Ruben Villegas**[1,4]    **Arkanath Pathak**[3]    **Harini Kannan**[2]
**Dumitru Erhan**[2]    **Quoc V. Le**[2]    **Honglak Lee**[2]
[1] University of Michigan
[2] Google Research
[3] Google
[4] Adobe Research

## Abstract

Predicting future video frames is extremely challenging, as there are many factors of variation that make up the dynamics of how frames change through time. Previously proposed solutions require complex inductive biases inside network architectures with highly specialized computation, including segmentation masks, optical flow, and foreground and background separation. In this work, we question if such handcrafted architectures are necessary and instead propose a different approach: finding minimal inductive bias for video prediction while maximizing network capacity. We investigate this question by performing the first large-scale empirical study and demonstrate state-of-the-art performance by learning large models on three different datasets: one for modeling object interactions, one for modeling human motion, and one for modeling car driving[1].

## 1 Introduction

From throwing a ball to driving a car, humans are very good at being able to interact with objects in the world and anticipate the results of their actions. Being able to teach agents to do the same has enormous possibilities for training intelligent agents capable of generalizing to many tasks. Model-based reinforcement learning is one such technique that seeks to do this – by first learning a model of the world, and then by planning with the learned model. There has been some recent success with training agents in this manner by first using video prediction to model the world. Particularly, video prediction models combined with simple planning algorithms [Hafner et al., 2019] or policy-based learning [Kaiser et al., 2019] for model-based reinforcement learning have been shown to perform equally or better than model-free methods with far less interactions with the environment. Additionally, Ebert et al. [2018] showed that video prediction methods are also useful for robotic control, especially with regards to specifying unstructured goal positions.

However, training an agent to accurately predict what will happen next is still an open problem. Video prediction, the task of generating future frames given context frames, is notoriously hard. There are many spatio-temporal factors of variation present in videos that make this problem very difficult for neural networks to model. Many methods have been proposed to tackle this problem [Oh et al., 2015, Finn et al., 2016, Vondrick et al., 2016, Villegas et al., 2017a, Lotter et al., 2017, Tulyakov et al., 2018, Liang et al., 2017, Denton and Birodkar, 2017, Wichers et al., 2018, Babaeizadeh et al., 2018, Denton and Fergus, 2018, Lee et al., 2018, Byeon et al., 2018, Yan et al., 2018, Kumar et al., 2018]. Most of these works propose some type of separation of information streams (e.g., motion/pose and content streams), specialized computations (e.g., warping, optical flow, foreground/background masks, predictive coding, etc), additional high-level information (e.g., landmarks, semantic segmentation masks, etc) or are simply shown to work in relatively simpler environments (e.g., Atari, synthetic shapes, centered human faces and bodies, etc).

Simply making neural networks larger has been shown to improve performance in many areas such as image classification [Real et al., 2018, Zoph et al., 2018, Huang et al., 2018], image generation [Brock et al., 2019], and language understanding [Devlin et al., 2018, Radford et al., 2019], amongst others. Particularly, Brock et al. [2019] recently showed that increasing the capacity of GANs [Goodfellow et al., 2014] results in dramatic improvements for image generation.

In his blog post "The Bitter Lesson", Rich Sutton comments on these types of developments by arguing that the most significant breakthroughs in machine learning have come from increasing the compute provided to simple models, rather than from specialized, handcrafted architectures [Sutton, 2019]. For example, he explains that the early specialized algorithms of computer vision (edge detection, SIFT features, etc.) gave way to larger but simpler convolutional neural networks. In this work, we seek to answer a similar question: do we really need specialized architectures for video prediction? Or is it sufficient to maximize network capacity on models with minimal inductive bias?

In this work, we perform the first large-scale empirical study of the effects of minimal inductive bias and maximal capacity on video prediction. We show that without the need of optical flow, segmentation masks, adversarial losses, landmarks, or any other forms of inductive bias, it is possible to generate high quality video by simply increasing the scale of computation. Overall, our experiments demonstrate that: (1) large models with minimal inductive bias tend to improve the performance both qualitatively and quantitatively, (2) recurrent models outperform non-recurrent models, and (3) stochastic models perform better than non-stochastic models, especially in the presence of uncertainty (e.g., videos with unknown action or control).

## 2 Related Work

The task of predicting multiple frames into the future has been studied for a few years now. Initially, many early methods tried to simply predict future frames in small videos or patches from large videos [Michalski et al., 2014, Ranzato et al., 2014, Srivastava et al., 2015]. This type of video prediction caused rectangular-shaped artifacts when attempting to fuse the predicted patches, since each predicted patch was blind to its surroundings. Then, action-conditioned video prediction models were built with the aim of being used for model-based reinforcement learning [Oh et al., 2015, Finn et al., 2016]. Later, video prediction models started becoming more complex and better at predicting future frames. Lotter et al. [2017] proposed a neural network based on predictive coding. Villegas et al. [2017a] proposed to separate motion and content streams in video input. Villegas et al. [2017b] proposed to predict future video as landmarks in the future and then use these landmarks to generate frames. Denton and Birodkar [2017] proposed to have a pose and content encoders as separate information streams. However, all of these methods focused on predicting a single future. Unfortunately, real-world video is highly stochastic – that is, there are multiple possible futures given a single past.

Many methods focusing on the stochastic nature of real-world videos have been recently proposed. Babaeizadeh et al. [2018] build on the optical flow method proposed by Finn et al. [2016] by introducing a variational approach to video prediction where the entire future is encoded into a posterior distribution that is used to sample latent variables. Lee et al. [2018] also build on optical flow and propose an adversarial version of stochastic video prediction where two discriminator networks are used to enable sharper frame prediction. Denton and Fergus [2018] also propose a similar variational approach. In their method, the latent variables are sampled from a prior distribution of the future during inference time, and only frames up to the current time step are used to model the future posterior distribution. Kumar et al. [2018] propose a method based on normalizing flows where the exact log-likelihood can be computed for training.

In this work, we investigate whether we can achieve high quality video predictions without the use of the previously mentioned techniques (optical flows, adversarial objectives, etc.) by just maximizing the capacity of a standard neural network. To the best of our knowledge, this work is the first to perform a thorough investigation on the effect of capacity increases for video prediction.

## 3 Scaling up video prediction

In this section, we present our method for scaling up video prediction networks. We first consider the Stochastic Video Generation (SVG) architecture presented in Denton and Fergus [2018], a stochastic video prediction model that is entirely made up of standard neural network layers without any special computations (e. g. optical flow). SVG is competitive with other state-of-the-art stochastic video

prediction models (SAVP, SV2P) [Lee et al., 2018]; however, unlike SAVP and SV2P, it does not use optical flow, adversarial losses, etc. As such, SVG was a fitting starting point to our investigation.

To build our baseline model, we start with the stochastic component that models the inherent uncertainty in future predictions from [Denton and Fergus [2018]. We also use shallower encoder-decoders that only have convolutional layers to enable more detailed image reconstruction [Dosovitskiy and Brox, 2016]. A slightly shallower encoder-decoder architecture results in less information lost in the latent state, as the resulting convolutional map from the bottlenecked layers is larger. Then, in contrast to Denton and Fergus [2018], we use a convolutional LSTM architecture, instead of a fully-connected LSTM, to fit the shallow encoders-decoders. Finally, the last difference is that we optimize the $\ell_1$ loss with respect to the ground-truth frame for all models like in the SAVP model, instead of using $\ell_2$ like in SVG. Lee et al. [2018] showed that $\ell_1$ encouraged sharper frame prediction over $\ell_2$.

We optimize our baseline architecture by maximizing the following variational lowerbound:

$$\sum_{t=1}^{T} \mathbb{E}_{q_\phi(\mathbf{z}_{\leq T}|\mathbf{x}_{\leq T})} \log p_\theta(\mathbf{x}_t|\mathbf{z}_{\leq t}, \mathbf{x}_{<t}) - \beta D_{KL} \left( q_\phi(\mathbf{z}_t|\mathbf{x}_{\leq t}) || p_\psi(\mathbf{z}_t|\mathbf{x}_{<t}) \right),$$

where $\mathbf{x}_t$ is the frame at time step $t$, $q_\phi(\mathbf{z}_{\leq T}|\mathbf{x}_{\leq T})$ the approximate posterior distribution, $p_\psi(\mathbf{z}_t|\mathbf{x}_{<t})$ is the prior distribution, $p_\theta(\mathbf{x}_t|\mathbf{z}_{\leq t}, \mathbf{x}_{<t})$ is the generative distribution, and $\beta$ regulates the strength of the KL term in the lowerbound. During training time, the frame prediction process at time step $t$ is as follows:

$$\mu_\phi(t), \sigma_\phi(t) = \text{LSTM}_\phi(\mathbf{h}_t; M) \qquad \text{where} \quad \mathbf{h}_t = f^{\text{enc}}(\mathbf{x}_t; K),$$
$$\mathbf{z}_t \sim \mathcal{N}(\mu_\phi(t), \sigma_\phi(t)),$$
$$\mathbf{g}_t = \text{LSTM}_\theta(\mathbf{h}_{t-1}, \mathbf{z}_t; M) \qquad \text{where} \quad \mathbf{h}_{t-1} = f^{\text{enc}}(\mathbf{x}_{t-1}; K),$$
$$\mathbf{x}_t = f^{\text{dec}}(\mathbf{g}_t; K),$$

where $f^{\text{enc}}$ is an image encoder and $f^{\text{dec}}$ is an image decoder neural network. $\text{LSTM}_\phi$ and $\text{LSTM}_\theta$ are LSTMs modeling the posterior and generative distributions, respectively. $\mu_\phi(t)$ and $\sigma_\phi(t)$ are the parameters of the posterior distribution modeling the Gaussian latent code $\mathbf{z}_t$. Finally, $\mathbf{x}_t$ is the predicted frame at time step $t$.

To increase the capacity of our baseline model, we use hyperparameters $K$ and $M$, which denote the factors by which the number of neurons in each layer of the encoder, decoder and LSTMs are increased. For example, if the number of neurons in LSTM is $d$, then we scale up by $d \times M$. The same applies to the encoder and decoder networks but using $K$ as the factor. In our experiments we increase both $K$ and $M$ together until we reach the device limits. Due to the LSTM having more parameters, we stop increasing the capacity of the LSTM at $M = 3$ but continue to increase $K$ up to 5. At test time, the same process is followed, however, the posterior distribution is replaced by the Gaussian parameters computed by the prior distribution:

$$\mu_\psi(t), \sigma_\psi(t) = \text{LSTM}_\psi(\mathbf{h}_{t-1}; M) \qquad \text{where} \quad \mathbf{h}_{t-1} = f^{\text{enc}}(\mathbf{x}_{t-1}; K),$$

Next, we perform ablative studies on our baseline architecture to better quantify exactly how much each individual component affects the quality of video prediction as capacity increases. First, we remove the stochastic component, leaving behind a fully deterministic architecture with just a CNN-based encoder-decoder and a convolutional LSTM. For this version, we simply disable the prior and posterior networks as described above. Finally, we remove the LSTM component, leaving behind only the encoder-decoder CNN architectures. For this version, we simply use $f^{\text{enc}}$ and $f^{\text{dec}}$ as the full video prediction network. However, we let $f^{\text{enc}}$ observe the same number of initial history as the recurrent counterparts.

Details of the devices we use to scale up computation can be found in the supplementary material.

## 4 Experiments

In this section, we evaluate our method on three different datasets, each with different challenges.

**Object interactions.** We use the action-conditioned towel pick dataset from Ebert et al. [2018] to evaluate how our models perform with standard object interactions. This dataset contains a robot arm that is interacting with towel objects. Even though this dataset uses action-conditioning, stochastic

| Dataset | CNN models | | LSTM models | | SVG' models | |
|---|---|---|---|---|---|---|
| | Biggest (M=3, K=5) | Baseline (M=1, K=1) | Biggest (M=3, K=5) | Baseline (M=1, K=1) | Biggest (M=3, K=5) | Baseline (M=1, K=1) |
| Towel Pick | 199.81 | 281.07 | 100.04 | 206.49 | **93.71** | 189.91 |
| Human 3.6M | 1321.23 | 1077.55 | 458.77 | 614.21 | **429.88** | 682.08 |
| KITTI | 2414.64 | 2906.71 | **1159.25** | 2502.69 | 1217.25 | 2264.91 |

Table 1: **Fréchet Video Distance evaluation (lower is better)**. We compare the biggest model we were able to train against the baseline models (M=1, K=1). Note that all models (SVG', CNN, and LSTM). The biggest recurrent models are significantly better than their small counterpart. Please refer to our supplementary material for plots showing how gradually increasing model capacity results in better performance.

video prediction is still required for this task. This is because the motion of the objects is not fully determined by the actions (the movements of the robot arm), but also includes factors such as friction and the objects' current state. For this dataset, we resize the original resolution of 48x64 to 64x64. For evaluation, we use the first 256 videos in the test set as defined by Ebert et al. [2018].

**Structured motion.** We use the Human 3.6M dataset [Ionescu et al., 2014] to measure the ability of our models to predict structured motion. This dataset is comprised of humans performing actions inside a room (walking around, sitting on a chair, etc.). Human motion is highly structured (i.e., many degrees of freedom), and so, it is difficult to model. We use the train/test split from Villegas et al. [2017b]. For this dataset, we resize the original resolution of 1000x1000 to 64x64.

**Partial observability.** We use the KITTI driving dataset [Geiger et al., 2013] to measure how our models perform in conditions of partial observability. This dataset contains driving scenes taken from a front camera view of a car driving in the city, residential neighborhoods, and on the road. The front view camera of the vehicle causes partial observability of the vehicle environment, which requires a model to generate seen and unseen areas when predicting future frames. We use the train/test split from Lotter et al. [2017] in our experiments. We extract 30 frame clips and skip every 5 frames from the test set so that the test videos do not significantly overlap, which gives us 148 test clips in the end. For this dataset, we resize the original resolution of 128x160 to 64x64.

### 4.1 Evaluation metrics

We perform a rigorous evaluation using five different metrics: Peak Signal-to-Noise Ratio (PSNR), Structural Similarity (SSIM), VGG Cosine Similarity, Fréchet Video Distance (FVD) [Unterthiner et al., 2018], and human evaluation from Amazon Mechanical Turk (AMT) workers. We perform these evaluations on all models described in Section 3: our baseline (denoted as SVG'), the recurrent deterministic model (denoted as LSTM), and the encoder-decoder CNN model (denoted as CNN). In addition, we present a study comparing the video prediction performance as a result of using skip-connections from every layer of the encoder to every layer of the decoder versus not using skip connections at all (Supplementary A.3), and the effects of the number of context frames (Supplementary A.4).

#### 4.1.1 Frame-wise evaluation

We use three different metrics to perform frame-wise evaluation: PSNR, SSIM, and VGG cosine similarity. PSNR and SSIM perform a pixel-wise comparison between the predicted frames and generated frames, effectively measuring if the exact pixels have been generated. VGG Cosine Similarity has been used in prior work [Lee et al., 2018] to compare frames in a perceptual level. VGGnet [Simonyan and Zisserman, 2015] is used to extract features from the predicted and ground-truth frames, and cosine similarity is performed at feature-level. Similar to Kumar et al. [2018], Babaeizadeh et al. [2018], Lee et al. [2018], we sample 100 future trajectories per video and pick the highest scoring trajectory as the main score.

#### 4.1.2 Dynamics-based evaluation

We use two different metrics to measure the overall realism of the generated videos: FVD and human evaluations. FVD, a recently proposed metric for video dynamics accuracy, uses a 3D CNN trained for video classification to extract a single feature vector from a video. Analogous to the well-known FID [Heusel et al., 2017], it compares the distribution of features extracted from ground-truth videos and generated videos. Intuitively, this metric compares the quality of the overall predicted video

| Dataset | LSTM models | | | SVG' models | | |
|---|---|---|---|---|---|---|
| | Biggest (M=3, K=5) | Baseline (M=1, K=1) | About the same | Biggest (M=3, K=5) | Baseline (M=1, K=1) | About the same |
| Towel Pick | **90.2**% | 9.0% | 0.8% | **68.8**% | 25.8% | 5.5% |
| Human 3.6M | **98.7**% | 1.3% | 0.0% | **95.8**% | 3.4% | 0.8% |
| KITTI | **99.3**% | 0.7% | 0.0% | **99.3**% | 0.7% | 0.0% |

Table 2: **Amazon Mechanical Turk human worker preference**. We compared the biggest and baseline models from LSTM and SVG'. The bigger models are more frequently preferred by humans. We present a full comparison for all large models in Supplementary A.5.

dynamics with that of the ground-truth videos rather than a per-frame comparison. For FVD, we also sample 100 future trajectories per video, but in contrast, all 100 trajectories are used in this evaluation metric (i.e., not just the max, as we did for VGG cosine similarity).

We also use Amazon Mechanical Turk (AMT) workers to perform human evaluations. The workers are presented with two videos (baseline and largest models) and asked to either select the more realistic video or mark that they look about the same. We choose the videos for both models by selecting the highest scoring videos in terms of the VGG cosine similarity with respect to the ground truth. We use 10 unique workers per video and choose the selection with the most votes as the final answer. Finally, we also show qualitative evaluations on pairs of videos, also selected by using the highest VGG cosine similarity scores for both the baseline and the largest model. We run the human perception based evaluation on the best two architectures we scale up.

### 4.2 Robot arm

For this dataset, we perform action-conditioned video prediction. We modify the baseline and large models to take in actions as additional input to the video prediction model. Action conditioning does not take away the inherent stochastic nature of video prediction due to the dynamics of the environment. During training time, the models are conditioned on 2 input frames and predict 10 frames into the future. During test time, the models predict 18 frames into the future.

**Dynamics-based evaluation.** We first evaluate the action-conditioned video prediction models using FVD to measure the realism in the dynamics. In Table 1 (top row), we present the results of scaling up the three models described in Section 3. Firstly, we see that our baseline architecture improves dramatically at the largest capacity we were able to train. Secondly, for our ablative experiments, we notice that larger capacity improves the performance of the vanilla CNN architecture. Interestingly, by increasing the capacity of the CNN architecture, it approaches the performance of the baseline SVG' architecture. However, as capacity increases, the lack of recurrence heavily affects the performance of the vanilla CNN architecture in comparison with the models that do have an LSTM (Supplementary A.2.1). Both the LSTM model and SVG' perform similarly well, with SVG' model performing slightly better. This makes sense as the deterministic LSTM model is more likely to produce videos closer to the ground truth; however, the stochastic component is still quite important as a good video prediction model must be both realistic and capable of handling multiple possible futures. Finally, we use human evaluations through Amazon Mechanical Turk to compare our biggest models with the corresponding baselines. We asked workers to focus on how realistic the interaction between the robot arm and objects looks. As shown in Table 2, the largest SVG' is preferred 68.8% of the time vs 25.8% of the time for the baseline (right), and the largest LSTM model is preferred 90.2% of the time vs 9.0% of the time for the baseline (left).

**Frame-wise evaluation.** Next, we use FVD to select the best models from CNN, LSTM, and SVG', and perform frame-wise evaluation on each of these three models. Since models that copy background pixels perfectly can perform well on these frame-wise evaluation metrics, in the supplementary material we also discuss a comparison against a simple baseline where the last observed frame is copied through time. From Figure 1, we can see that the CNN model performs much worse than the models that have recurrent connections. This is a clear indication that recurrence is necessary to predict future frames, and capacity cannot make up for it. Both LSTM and SVG perform similarly well, however, towards the end, SVG slightly outperforms LSTM. The full evaluation on all capacities for SVG', LSTM, and CNN is presented in the supplementary material.

**Qualitative evaluation.** In Figure 2, we show example videos from the smallest SVG' model, the largest SVG' model, and the ground truth. The predictions from the small baseline model are blurrier

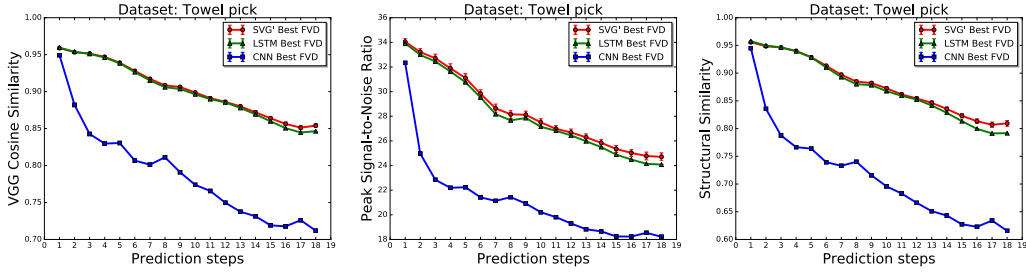

Figure 1: Towel pick per-frame evaluation (higher is better). We compare the best performing models in terms of FVD. For model capacity comparisons, please refer to Supplementary A.2.1.

compared to the largest model, while the edges of objects from the larger model's predictions stay continuously sharp throughout the entire video. This is clear evidence that increasing the model capacity enables more accurate modeling of the pick up dynamics. For more videos, please visit our website `https://cutt.ly/QGuCex`.

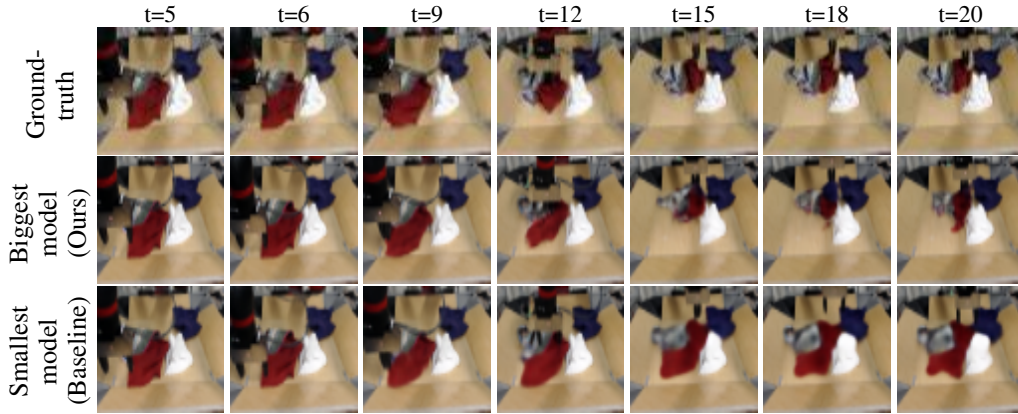

Figure 2: Robot towel pick qualitative evaluation. Our highest capacity model (middle row) produces better modeling of the robot arm dynamics, as well, as object interactions. The baseline model (bottom row) fails at modeling the objects (object blurriness), and also, the robot arm dynamics are not well modeled (gripper is open when the it should be close at t=18). For best viewing and more results, please visit our website `https://cutt.ly/QGuCex`.

### 4.3 Human activities

For this dataset, we perform action-free video prediction. We use a single model to predict all action sequences in the Human 3.6M dataset. During training time, the models are conditioned on 5 input frames and predict 10 frames into the future. At test time, the models predict 25 frames.

**Dynamics-based evaluation.** We evaluate the predicted human motion with FVD (Table 1, middle row). The performance of the CNN model is poor in this dataset, and increasing the capacity of the CNN does not lead to any increase in performance. We hypothesize that this is because the lack of action conditioning and the many degrees of freedom in human motion makes it very difficult to model with a simple encoder-decoder CNN. However, after adding recurrence, both LSTM and SVG' perform significantly better, and both models' performance become better as their capacity is increased (Supplementary A.2.2). Similar to Section 4.2, we see that SVG' performs better than LSTM. This is again likely due to the ability to sample multiple futures, leading to a higher probability of matching the ground truth future. Secondly, in our human evaluations for SVG', 95.8% of the AMT workers agree that the bigger model has more realistic videos in comparison to the smaller model, and for LSTM, 98.7% of the workers agree that the LSTM largest model is more realistic. Our results, especially the strong agreement from our human evaluations, show that high capacity models are better equipped to handle the complex structured dynamics in human videos.

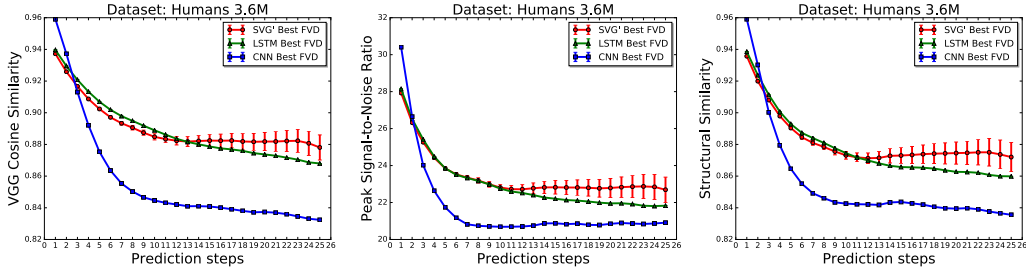

Figure 3: Human 3.6M per-frame evaluation (higher is better). We compare the best performing models in terms of FVD. For model capacity comparisons, please refer to Supplementary A.2.2.

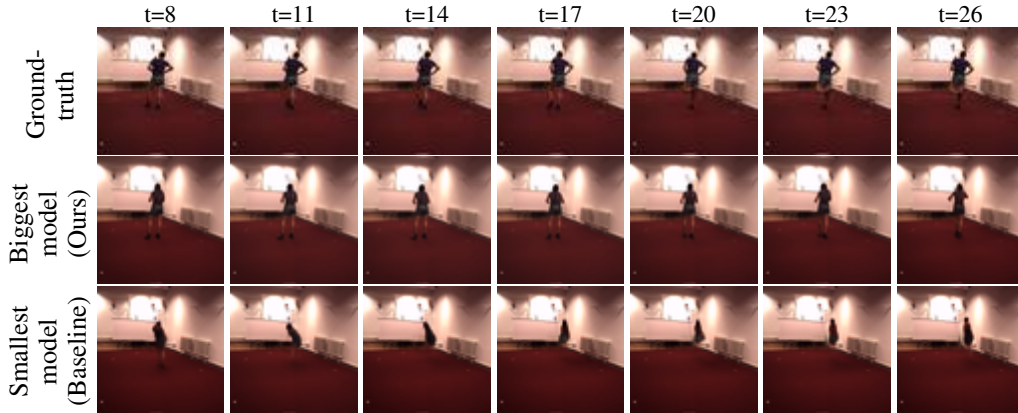

Figure 4: Human 3.6M qualitative evaluation. Our highest capacity model (middle) produces better modeling of the human dynamics. The baseline model (bottom) is able to keep the human dynamics to some degree but in often cases the human shape is unrecognizable or constantly vanishing and reappearing. For more videos, please visit our website `https://cutt.ly/QGuCex`.

**Frame-wise evaluation.**   Similar to the previous per-frame evaluation, we select the best performing models in terms of FVD and perform a frame-wise evaluation. In Figure 3, we can see that the CNN based model performs poorly against the LSTM and SVG' baselines. The recurrent connections in LSTM and SVG' are necessary to be able to identify the human structure and the action being performed in the input frames. In contrast to Section 4.2, there are no action inputs to guide the video prediction which significantly affects the CNN baseline. The LSTM and SVG' networks perform similarly at the beginning of the video while SVG' outperforms LSTM in the last time steps. This is a result of SVG' being able to model multiple futures from which we pick the best future for evaluation as described in Section 4.1. We present the full evaluation on all capacities for SVG', LSTM, and CNN in the supplementary material.

**Qualitative evaluation.**   Figure 4 shows a comparison between the smallest and largest stochastic models. In the video generated by the smallest model, the shape of the human is not well-defined at all, while the largest model is able to clearly depict the arms and the legs of the human. Moreover, our large model is able to successfully predict the human's movement throughout all of the frames into the future. The predicted motion is close to the ground-truth motion providing evidence that being able to model more factors of variation with larger capacity models can enable accurate motion identification and prediction. For more videos, please visit our website `https://cutt.ly/QGuCex`.

## 4.4   Car driving

For this dataset, we also perform action-free video prediction. During training time, the models are conditioned on 5 input frames and predict 10 frames into the future. At test time, the models predict 25 frames into the future. This video type is the most difficult to predict since it requires the model to be able to hallucinate unseen parts in the video given the observed parts.

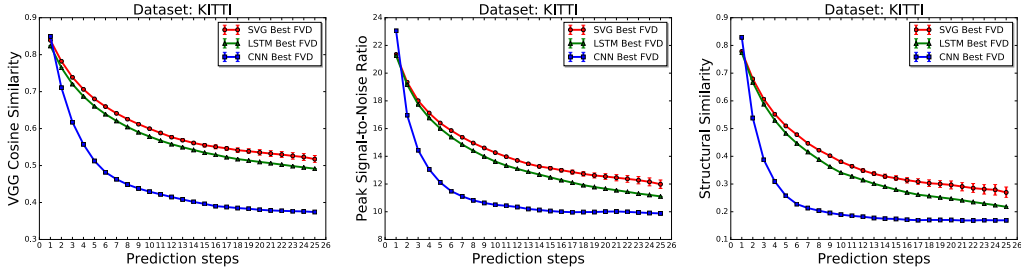

Figure 5: KITTI driving per-frame evaluation (higher is better). For model capacity comparisons, please refer to Supplementary A.2.3.

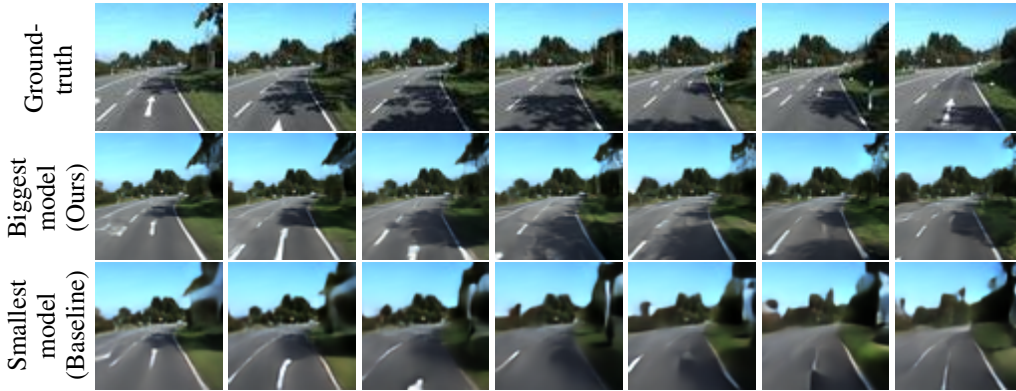

Figure 6: KITTI driving qualitative evaluation. Our highest capacity model (middle) is able to maintain the observed dynamics of driving forward and is able to generate unseen street lines and the moving background. The baseline (bottom) loses the street lines and the background becomes blurry. For best viewing and more results, please visit our website `https://cutt.ly/QGuCex`.

**Dynamics-based evaluation.** We see very similar results to the previous dataset when measuring the realism of the videos. For both LSTM and SVG', we see a large improvement in FVD when comparing the baseline model to the largest model we were able to train (Table 1, bottom row). However, we see a similarly poor performance for the CNN architecture as in Section 4.3, where capacity does not help. One interesting thing to note is that the largest LSTM model performs better than the largest SVG' model. This is likely related to the architecture design and the data itself. The movements of cars driving is mostly predictable, and so, the deterministic architecture becomes highly competitive as we increase the model capacity (Supplementary A.2.3). However, our original premise that increasing model's capacity improves network performance still holds. Finally, for human evaluations, we see in Table 2 that the largest capacity SVG' model is preferred by human raters 99.3% of the time (right), and the largest capacity LSTM model (left) is also preferred by human raters 99.3% time (left).

**Frame-wise evaluation** Now, when we evaluate based on frame-wise accuracy, we see similar but not exactly the same behavior as the experiments in Section 4.3. The CNN architecture performs poorly as expected, however, LSTM and SVG' perform similarly well.

**Qualitative evaluation.** In Figure 6, we show a comparison between the largest stochastic model and its baseline. The baseline model starts becoming blurry as the predictions move forward in the future, and important features like the lane markings disappear. However, our biggest capacity model makes very sharp predictions that look realistic in comparison to the ground-truth.

## 5 Higher resolution videos

Finally, we experiment with higher resolution videos. We train SVG' on the Human 3.6M and KITTI driving datasets. These two datasets contain much larger resolution images compared to the

Towel pick dataset, enabling us to sub-sample frames to twice the resolution of previous experiments (128x128). We follow the same protocol for the number of input and predicted time steps during training (5 inputs and 10 predictions), and the same protocol for testing (5 inputs and 25 predictions). In contrast to the networks used in the previous experiments, we add three more convolutional layers plus pooling to subsample the input to the same convolutional encoder output resolution as in previous experiments.

In Figure 7 we show qualitative results comparing the smallest (baseline) and biggest (Ours) networks. The biggest network we were able to train had a configuration of M=3 and K=3. Higher resolution videos contain more details about the pixel dynamics observed in the frames. This enables the models to have a denser signal, and so, the generated videos become more difficult to distinguish from real videos. Therefore, this result suggests that besides training better and bigger models, we should also more towards larger resolutions. For more examples of videos, please visit our website: `https://cutt.ly/QGuCex`.

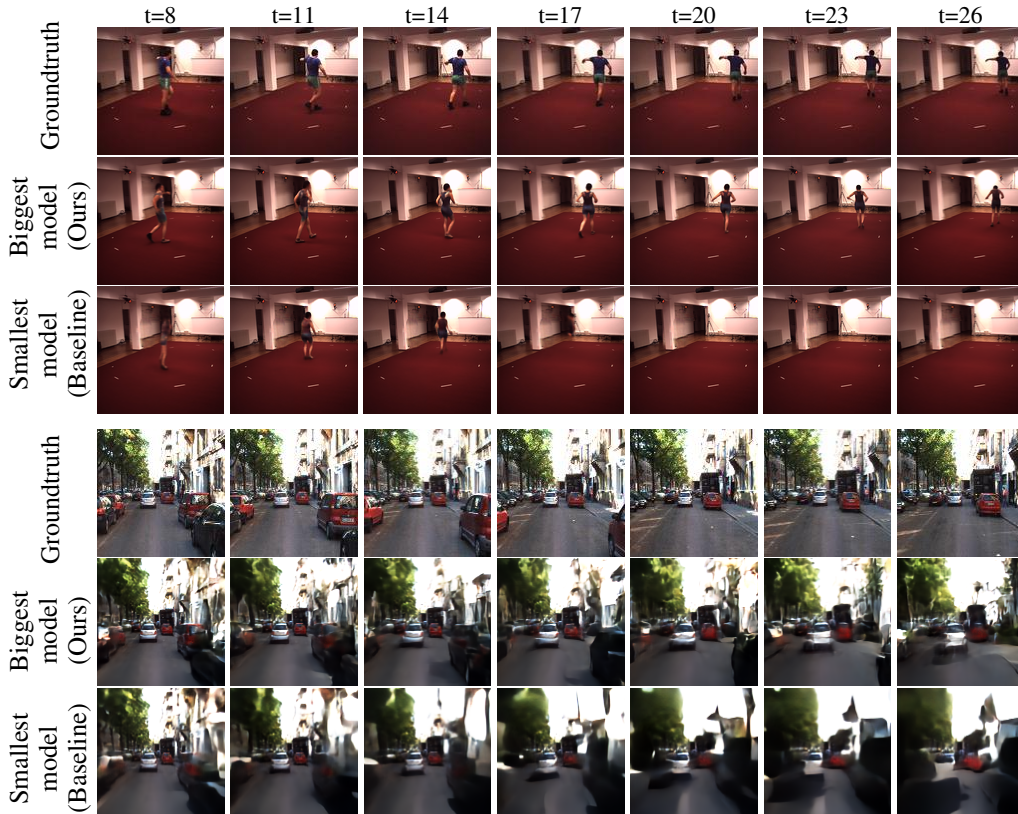

Figure 7: Human 3.6M and KITTI driving qualitative evaluation on high resolution videos (frame size of 128x128) with comparison between smallest model and largest model we were able to train (M=3, K=3). For best viewing and more results, please visit our website `https://cutt.ly/QGuCex`.

# 6   Conclusion

In conclusion, we provide a full empirical study on the effect of finding minimal inductive bias and increasing model capacity for video generation. We perform a rigorous evaluation with five different metrics to analyze which types of inductive bias are important for generating accurate video dynamics, when combined with large model capacity. Our experiments confirm the importance of recurrent connections and modeling stochasticity in the presence of uncertainty (e.g., videos with unknown action or control). We also find that maximizing the capacity of such models improves the quality of video prediction. We hope our work encourages the field to push along similar directions in the future – i.e., to see how far we can get by finding the right combination of minimal inductive bias and maximal model capacity for achieving high quality video prediction.

## Footnotes

[1]This work was done while the first author was an intern at Google

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
