[Supplementary Material]

# A Supplementary material

## A.1 Video results

We have provided video comparisons of the baseline and largest model for the best two models (LSTM and SVG') in this website: `https://cutt.ly/QGuCex`.

## A.2 Per-frame evaluation comparison as model capacity increases

In this section, we present a per-frame evaluation for capacities in each of the models we experiment in our paper.

### A.2.1 Robot arm.

The plots show a slight improvement as the number of parameters increase for the CNN architecture. However, for the LSTM and SVG' architectures the improvement is more noticeable. We hypothesize that this is due to the model being able to better handle the robot arm interaction with the objects by having a large capacity.

Figure 8: Towel pick per-frame evaluation (higher is better). As capacity increases, the per frame evaluation metrics become better. The increase is due to better modeling of interactions. The objects become sharper, and robot arm dynamics become better as the model capacity increases.

### A.2.2 Human activities.

The Human 3.6M dataset is mostly made of static background and the moving human occupies a relatively very small area of the frame. Therefore, models that are not capable of perfectly predicting the background become affected by this. To show our point, we include a baseline where we simply copy the last observed frame through time. This baseline significantly outperforms all models. Therefore, from these results we can conclude that per-frame evaluations are not reliable when a large portion of a video does not move.

Figure 9: Human 3.6M per-frame evaluation (higher is better). In this dataset, there is a large amount of non-moving background that causes a per-frame evaluation to become not reliable. This is shown by the baseline based on simply copying the last observed frame through time which significantly outperforms all methods.

### A.2.3 Car driving.

In this dataset, as observed by the FVD measure in the main text, we see that the CNN model fails to make improvement in the per-frame evaluation metrics. However, the LSTM and SVG' models performance improves as the capacity of the models increases. The metric in which this is the most obvious is the VGG Cosine Similarity. This may be due to the partial observability of the dataset which makes it very difficult to predict exact pixels into the future, and so, PSNR and SSIM do not result in a large gap between the larger and baseline models. However, VGG Cosine Similarity compares high-level features of the predicted frames. Therefore, even if the predicted pixels are not exact, the predicted structures in the frames may be similar to those the ground-truth future. For this dataset, we do not present a copy last frame baseline because most pixels move (in contrast to the robot arm and Human 3.6M dataset, where many pixels stay fixed).

Figure 10: KITTI driving per-frame evaluation (higher is better). As capacity increases, the per frame evaluation metrics become better. The increase is due to better modeling the driving dynamics and partial observability. Due to the difficulty of predicting the exact not-observed parts of the image, the performance converges toward the largest models.

### A.3 Effects of using skip connections in video prediction

In this section, we present a study on the effects of using skip connections from encoder to decoder. Similar to Denton and Fergus [2018], the method presented in the main text has skip connections going from the encoder of the last observed frame directly to the decoder for all frame predictions. This allows the video prediction method to choose to transfer pixels that did not move from the input frame directly into the output frame, and generate the pixels that move. Below, we show the performance for each of the datasets presented in this work.

#### A.3.1 Robot Arm.

In Figure 11, we can see that skip connections do play an important role in terms of FVD evaluation for the robot arm action conditioned experiments. This implies that having skip connections eases the difficulty of video prediction in that it is only required to model the dynamics of the moving parts and everything else can simply be transferred to the output frames.

Figure 11: Towel pick video dynamics evaluation (lower is better). Solid lines define method with skip connections and dotted lines without skip connections.

In addition, having skip connections also help to make more accurate frame-wise predictions. In Figure 12, the advantage of having skip connections is clear in all prediction steps. This indicates that skip connections are not just essential for predicting dynamics that look like the ground-truth videos, but also, the accuracy of the predicted pixels becomes better.

Figure 12: Towel pick per-frame evaluation (higher is better). Solid lines define method with skip connections and dotted lines without skip connections.

#### A.3.2 Human activities.

In Figure 13, having skip connections results in a large performance improvement in FVD for the CNN based video prediction architecture. However, for the LSTM and SVG' based architectures, we can that there is not clear improvement as the model size increases. We hypothesize that, since there are no interactions, the background is static, and the background between training and testing data is similar, the dataset dynamics become easier to model. Therefore, there is no need for the model to separate moving and non-moving parts to achieve good predictions.

Figure 13: Human 3.6M video dynamics evaluation (lower is better). Solid lines define method with skip connections and dotted lines without skip connections.

In contrast to FVD evaluation, having skip connections greatly improves the performance in the per-frame evaluation metrics for all models (Figure 14). This is mainly due to the fact that the moving humans take up a very small portion of the image. Thus, having a way to transfer non-moving pixels directly into the output frames results in more accurate per-frame performance.

Figure 14: Human 3.6M per-frame evaluation (higher is better). Solid lines define method with skip connections and dotted lines without skip connections.

### A.3.3    KITTI driving.

In Figure 15, we can see that for the recurrent models (LSTM and SVG') having skip connections results in improved FVD performance. However, when using a CNN based architecture, is clear for most models, but not all of them as the two curves become close to each other when $M$ and $K$ are make the model twice and three times bigger than the original model (second and third parameter value in the x-axis). We hypothesize that this happens because almost all pixels move in these videos, and so, simple skip connections without recurrent steps to remember what pixels are moving throughout the prediction makes skip connections not as critical for the intermediate size models.

Figure 15: KITTI driving video dynamics evaluation (lower is better). Solid lines define method with skip connections and dotted lines without skip connections.

In terms of per-frame evaluation, we see an interesting behavior as prediction move forward in time (Figure 16). The predicted frames become less accurate as time moves forward; effectively reducing the performance gap between the architectures with and without skip connections. This happens because predicting videos in this dataset requires predicting unseen pixels moving into view (e.g., partial observability). Therefore, having skip connections can only help for predicting nearby frames and eventually requires generating fully unseen objects in the frames. The probability that the exact pixels are generated reduces as time moves forward, even if the overall predicted dynamics are within what is realistic in the dataset.

Figure 16: KITTI driving per-frame evaluation (higher is better). Solid lines define method with skip connections and dotted lines without skip connections.

## A.4 Effects of the number of context frames

In this section, we present a study over number of context frames given to each of the considered networks. We consider models that observe 2, 5 and 10 frames to predict 20 frames into the future for our action-free experiments (Human 3.6M and KITTI), and models observe 2, 4 and 8 frames to predict 12 frames into the future for our action-conditioned experiments (Towel pick). We test on a slightly different test set from the one in the main paper to make sure the future frames during evaluation are all the same for all the models in this section. We present the per-frame metrics used in the main paper but averaged over time, and also, the Fréchet Video Distance (FVD) dynamics evaluation metric.

### A.4.1 Per-frame evaluation

Firstly, we perform per-frame evaluation of the predicted frames. We want to observe how context affects the accuracy of the predicted future with respect to the ground-truth future.

In Table 3 (action-free evaluation), we can see that increasing the number of context frames improves the performance in most of the recurrent models or converges at context of 5 frames. In contrast, we cannot conclude the same for the CNN models. In fact, most of our experiments perform better with less number of context frames. We hypothesize that this may be due to the lack of recurrence in the CNN model which has to infer dynamics from all context frames in one shot at every prediction step while not keeping a history. The recurrent models have the advantage of keeping a history while deciding what information to keep or discard.

| Dataset | Metric | Network | Context = 2 | Context = 5 | Context = 10 |
|---------|--------|---------|-------------|-------------|--------------|
| Human 3.6M | Cosine Sim. | SVG | 0.916 | **0.925** | **0.925** |
| | | LSTM | 0.912 | 0.918 | **0.921** |
| | | CNN | **0.890** | 0.869 | 0.875 |
| | PSNR | SVG | 23.420 | 23.778 | **23.948** |
| | | LSTM | 22.956 | 23.391 | **23.734** |
| | | CNN | **22.804** | 21.740 | 21.845 |
| | SSIM | SVG | 0.880 | 0.889 | **0.892** |
| | | LSTM | 0.876 | 0.883 | **0.886** |
| | | CNN | 0.862 | 0.857 | **0.863** |
| KITTI | Cosine Sim. | SVG | 0.689 | **0.702** | 0.700 |
| | | LSTM | 0.639 | 0.672 | **0.688** |
| | | CNN | **0.594** | 0.493 | 0.508 |
| | PSNR | SVG | 14.549 | 14.953 | **14.960** |
| | | LSTM | 13.623 | 14.476 | **14.694** |
| | | CNN | **13.522** | 11.883 | 11.989 |
| | SSIM | SVG | 0.403 | **0.419** | 0.417 |
| | | LSTM | 0.349 | 0.387 | **0.403** |
| | | CNN | **0.316** | 0.264 | 0.275 |

Table 3: Average per-frame evaluation of the effects of the number of context frames in the action-free datasets (Human 3.6M and KITTI). We compare models with different number of context frames and prediction of 20 frames.

In Table 4 (action-conditioned evaluation), we see a similar pattern as in Table 3 for the recurrent models. Having more context frames enables recurrent models to make more accurate predictions of the future with respect to the ground-truth future. In addition, the CNN based architecture performance does not degrade as more context frames are given as input. Having actions as input makes the prediction easier, and the CNN does not have to infer all future frame dynamics from pixels alone.

| Dataset | Metric | Network | Context = 2 | Context = 4 | Context = 8 |
|---------|--------|---------|-------------|-------------|-------------|
|  | | SVG | 0.906 | 0.926 | **0.932** |
|  | Cosine Sim. | LSTM | 0.904 | 0.922 | **0.931** |
|  | | CNN | 0.835 | 0.819 | **0.837** |
|  | | SVG | 26.125 | 27.814 | **28.703** |
| Towel pick | PSNR | LSTM | 25.328 | 27.304 | **28.706** |
|  | | CNN | 21.425 | 20.913 | **21.767** |
|  | | SVG | 0.834 | 0.868 | **0.875** |
|  | SSIM | LSTM | 0.827 | 0.862 | **0.878** |
|  | | CNN | 0.725 | 0.708 | **0.729** |

Table 4: Average per-frame evaluation of the effects of the number of context frames in the action-conditioned datasets (Towel pick). We compare models with different number of context frames and prediction of 12 frames.

### A.4.2 Fréchet Video Distance evaluation

In this section, we evaluate the dynamics of the generated videos using the Fréchet Video Distance (FVD). In Table 5, we see a similar pattern in the Human 3.6M and KITTI driving experiments. For the SVG architecture, 5 context frames are the most optimal number of frames in terms to predict the best full video dynamics. In the LSTM architecture, 10 context frames are the most optimal. Finally, for the CNN architecture, 2 context frames are the most optimal. From these results, we see that for both datasets the SVG model the improvement stops at 5 context frames. This could be due to the more conditioning frames impacting the predictions in terms of the distribution of future dynamics. However, we need to investigate further to determine why this is happening. For the LSTM model, more context frames keep improving the predicted dynamics quality. Finally, for the CNN architecture, we see a similar behavior as in the per-frame evaluations where less context frames are better for inferring future dynamics.

| Dataset | Metric | Network | Context = 2 | Context = 5 | Context = 10 |
|---------|--------|---------|-------------|-------------|--------------|
|  | | SVG | 440.511 | **428.792** | 434.743 |
| Human 3.6M | FVD | LSTM | 484.011 | 490.375 | **463.984** |
|  | | CNN | **470.751** | 1006.216 | 908.939 |
|  | | SVG | 1183.945 | **1125.285** | 1391.642 |
| KITTI | FVD | LSTM | 1309.101 | 1228.919 | **1224.859** |
|  | | CNN | **1408.143** | 2673.012 | 2494.317 |

Table 5: Fréchet Video Distance (FVD) evaluation of the effects of the number of context frames in the action-free datasets (Human 3.6M and KITTI). We compare models with different number of context frames and prediction of 20 frames.

In Table 6, we see a slightly different result in comparison to Table 5. For both SVG and LSTM architectures, 8 context frames (the most we tried) are the most optimal number of frames in terms to predict the best video dynamics. The difference in these experiments is that we have action inputs that determine the robot arm motion (albeit the objects with which the arm interacts still have a stochastic behavior). For the CNN architecture, 2 context frames are the most optimal. This is the same finding we have in table 5 for both action-free datasets regarding the predicted video dynamics.

| Dataset | Metric | Network | Context = 2 | Context = 4 | Context = 8 |
|---------|--------|---------|-------------|-------------|-------------|
| Towel Pick | FVD | SVG | 93.977 | 71.415 | **69.038** |
| | | LSTM | 96.138 | 73.494 | **67.015** |
| | | CNN | **127.281** | 143.394 | 131.376 |

Table 6: Fréchet Video Distance (FVD) evaluation of the effects of the number of context frames in the action-conditioned dataset (Towel Pick). We compare models with different number of context frames and prediction of 12 frames.

## A.5 All-vs-all Amazon Mechanical Turk comparison

In this section, we compare the largest models we trained for the different inductive bias considered in our study. Similar to the experiments presented in the may text, we use 10 unique workers per video and choose the selection with the most votes as the final answer. The videos used in the comparison are determined by the highest VGG Cosine Similarity score amongst all samples for the stochastic model, and we use the single trajectory produced by LSTM and CNN.

| Dataset | Method 1 | Method 2 | Method 1 | Method 2 | About the same |
|---------|----------|----------|----------|----------|----------------|
| Towel Pick | SVG | LSTM | 43.8% | 53.5% | 2.7% |
| | SVG | CNN | 38.7% | 58.2 % | 3.1% |
| | CNN | LSTM | 32.7% | 66.0% | 2.0% |
| Human 3.6M | SVG | LSTM | 34.5% | 63.0% | 2.5% |
| | SVG | CNN | 96.6% | 2.9% | 0.4% |
| | CNN | LSTM | 2.5% | 97.5% | 0.0% |
| KITTI | SVG | LSTM | 55.4% | 44.6% | 0.0% |
| | SVG | CNN | 97.3% | 2.7% | 0.0% |
| | CNN | LSTM | 0.7% | 99.3% | 0.0% |

Table 7: **Amazon Mechanical Turk human worker preference**. We compared the biggest and baseline models from LSTM and SVG'. The bigger models are more frequently preferred by humans.

### A.6 Device and network details

To scale up the capacity of the model, we use 32 Google TPUv3 Pods [Google, 2018] for each experiment and a batch size of 32. We distribute the training batch such that there is a single batch element in each 16GB TPU. This way we can use each device to the maximum capacity. We first increase $K$ and $M$ together while keeping $K$ to be equals to $M$. By simply doubling the number of neurons in each layer, we see an improvement. We then continue to increase $K$ and $M$ up to three times the number of neurons in each layer. At this, point we are not able to increase $M$ anymore without running out of memory, and so, we only continue increasing $K$.

### A.7 Architecture and hyper-parameters

For the encoder network we use VGG-net [Simonyan and Zisserman, 2015] up to layer conv3_3 after pooling and a single convolutional layer with output of 128 channels. A mirrored architecture of the encoder is used for the decoder network. For the Convolutional LSTMs used throughout we use a single layer network with 512 units for $\text{LSTM}_\psi$ and $\text{LSTM}_\phi$, and a two layer network with 512 units for $\text{LSTM}_\theta$. Other than that, we follow a similar architecture as Denton and Fergus [2018] including the skip connections from encoder to decoder. We use $\beta = 0.0001$ for all of our experiments. The number of hidden units in $z$ are 64 for the robot arm dataset and 128 for all other datasets.