[Reviews · NeurIPS 2019]

Reviewer 1



Quality: The paper is technically sound. Claims are supported by experimental results. The experimental study tests the proposed method on three standard datasets with correct methodologies and evaluations. Model capacity comparisons are covered in the supplementary material. I believe those comparisons are important, and it is better if authors can include them in the main paper. The paper tries to show that handcrafted features, separation of information sources, or any other specialized computation are not necessary for the video prediction task. However, the experimental results are only comparing the performance of simple networks with various capacities. This only shows that more capacity results in better results but It is not clear how other SOTA methods are compared to the simple architectures. Varying the capacity of SOTA networks, or simply reporting their performance can better evaluate the claim. Clarity: The paper is generally well-written and structured clearly. Originality: To my knowledge, there is no other paper providing a large scale study of networks capacity on the task of video prediction. The literature review seems to be complete. Significance: The results of this paper are important for research in processing videos. It is somewhat expected (from observing the same phenomena in other tasks) that higher capacity results in better performances, but this expectation has never been proved before. This paper is a useful contribution to the literature and has high practical impact. The paper may be better suited for publication in CVPR or ICLR conferences or perhaps can be published as an application paper at NIPS. Update: I think this paper is a good contribution to the field and might be helpful for other video processing tasks as well. That being said, I do believe comparing to the performance of SOTA networks is important. The author response mentions that a SOTA network has been only evaluated on simpler datasets. I do encourage the authors to compare to SOTA on the main datasets of the paper, but if the computation and hyperparameter tuning is really impractical, it is essential to at least have a comparison and report the performance of standard networks with different capacities on the datasets that SOTA is already evaluated on.

Reviewer 2



This paper attempts to answer the question of whether it is necessary to have the specialized architectures to solve the video prediction problem, and hypothesizes that the specialized architectures are not necessary and instead “standard” neural networks are capable of solving the problem as long as the networks are large enough. The paper then goes on in verifying this hypothesis using “standard” large networks in three datasets corresponding to three different types of video activities for video prediction. The paper claims that this was the first work to answer the question of whether it is necessary to specialized architectures for video prediction and provides the empirical study addressing this question with “standard” large networks. While I am not sure whether this was the first such work on this question (I personally did not see such work before indeed), the hypothesis is pretty much in line with the well known understanding that larger networks may provide better generalization capability in general, that is supported in recent literature including those mentioned in the related work in this paper. Consequently, I am not surprised to see that the hypothesis was verified in the empirical studies reported in the paper, and thus, I don’t see that the novelty of this work is terribly impressive though I appreciate the efforts in conducting and reporting the experiments in this paper. Regarding the empirical studies reported in the paper, I have the following two more comments/requests. First, for the scenes in the three datasets studied (object interaction, human motion, and car driving), they are relatively causality-explicit in nature, and thus there is not much uncertainty in prediction. What if you have a scene with more uncertainty or stochasticity such as natural scene motion (e.g., tornado) or events (e.g., crowd gathering, earthquake)? Second, presumably, the prediction accuracy also depends upon the length of the history data. Can you provide such empirical studies in the three datasets? The above was my original review. I read authors' response and am happy with their answers. I still have my reservation regarding the novelty issue. But I am happy to bump up the overall score to 7.

Reviewer 3



Edit: I read the author rebuttal and the other reviews. I will not change my score. I still encourage the authors to try adding experiments that vary the capacity of the SAVP approach, even if it is computationally difficult. We should explore and understand how the performance of both traditional approaches and more complicated approaches change with varying capacity. -------------- Originality -It is not a new concept that high capacity models can improve prediction tasks, but the authors apply the idea to video prediction and are the first to demonstrate that such models can perform well on this task. -The paper may also be the first to perform such thorough and rigorous evaluation (e.g. a study across three datasets and five metrics) of models at varying capacity levels for this task. -Related work is extremely well cited throughout the paper. Quality -This paper is an empirical study of architecture design, and a strength of the work is indeed empirical rigor. The authors propose five metrics for evaluation and compare on three datasets. The type of models that are tested are clearly thoughtfully chosen based on a careful literature review. -The authors include human evaluation using AMT, which, in my opinion, greatly increases the quality and usefulness of the evaluations -One critique is that there is no direct comparison to some of the techniques that the authors claim are not needed for this task. It would have been nice to see how the performance of architectures that employ optical flow, segmentation tasks, adversarial losses, and landmarks scale with increasing capacity. Did the authors try this? (it is possible I just missed something about the paper). Clarity -The paper was well written and easy to follow. Significance The idea that high capacity models can perform well compared to specialized, handcrafted architectures is not novel, as the authors point out in the introduction. However, this paper is the first to apply the idea to video prediction and the first to conduct a rigorous experimental study of this type. So, in my opinion, the results are somewhere in the range of a medium to high level of significance. If the idea were completely new, the paper would be more significant to me, but I do think that this work will help guide researchers to design better architectures for this task.

[Author Response · NeurIPS 2019]

We thank the reviewers for their insightful and constructive comments.

**[R3] "... larger networks provide better generalization":** We not only show that larger networks provide better generalization capacity per se, but also provide rigorous studies about (1) the minimal inductive bias necessary to achieve high quality video prediction, (2) quantifying the gains resulting from each architectural component, and (3) quantifying the gains resulting from gradual increase in capacity. We show the progression and performance increase going from an encoder/decoder CNN (CNN); to adding a recurrent component (LSTM); to adding a recurrent stochastic component in the architecture (SVG). In addition, our study progressively increases the difficulty of the datasets to highlight how each of the models being studied perform at each level of difficulty (i.e., action conditioned prediction, action-free with static background, action-free with moving background). As highlighted by R1 and R4, although the idea that increasing capacity can be beneficial for model performance may not be a surprise, our paper is the first to successfully and comprehensively demonstrate and quantify this for video prediction over five different metrics. A lot of effort goes into discovering domain-specific architectures (e.g. using optical flow, segmentation masks, and other forms of inductive bias) – and we hope our work encourages the field to rethink about these aspects of scalability.

**[R3] "Datasets are relatively causality-explicit, and thus, not much uncertainty in prediction":** While we agree that action conditioning limits the uncertainty for the BAIR experiments, there is still partial observability in the object interactions. The model has to hallucinate the unseen parts of the objects and also any stochasticity in the interaction which cannot be fully determined by observing the pixels (e.g., table friction). On the other hand, Human 3.6M and KITTI contain larger amounts of stochasticity. First, the actions in the Human 3.6M dataset are highly stochastic, that is, the human randomly decides to do different actions regardless of the label (e.g., "sitting" action randomly goes from sitting to getting up to walking). This makes the prediction not fully determined by the observations so the model has to choose one of the possible futures and predict it. Second, the driving data from KITTI is also highly stochastic due to strong partial observability. Given input frames from the driving scene, models need to be able to hallucinate the road and vehicles that are hidden by the horizon line caused. Having said that, we also agree that it is interesting to evaluate on datasets with higher uncertainties (though not as well established in the video prediction literature) and will try to include such results in the final version.

**[R3] Prediction accuracy depending on the context length:** We ran experiments with history length of 5 and 10 frames. We evaluated with the same data as in the submission (30 frames total), thus, we evaluate by predicting 20 frames into the future so we can align the future frames for comparison. Due to space limitations, we cannot provide full sequence plots for the frame-wise evaluation, and so, we provide the average over all time steps. Also, due to time constraints, we trained the baseline (smallest) model with M=1 and K=1. We will add results for the biggest models in the final version. For similar reasons, we couldn't run experiments on the robot dataset. However, since there is action conditioning on the robot dataset, context frames may be less influential. Overall, we observe that most of the metrics improve with more context frames—i.e., 7 out of 8 evaluation settings except for the case of FVD on Human3.6M (each row in the table corresponds to a combination of evaluation metric and dataset). We further expect that larger-sized models will perform better with longer context size and will report more comprehensive results in the final version.

| | | CNN models | | LSTM models | | SVG' models | |
|---|---|---|---|---|---|---|---|
| Dataset | Metric | history=5 | history=10 | history=5 | history=10 | history=5 | history=10 |
| Human 3.6M | PSNR **(higher/better)** | 22.351 | 22.522 | 22.927 | 23.108 | 22.841 | **23.399** |
| | SSIM **(higher/better)** | 0.873 | 0.877 | 0.886 | **0.894** | 0.887 | 0.891 |
| | Cos. Sim. **(higher/better)** | 0.882 | 0.881 | 0.898 | **0.903** | 0.899 | 0.902 |
| | FVD **(lower/better)** | 848.714 | 890.270 | 616.474 | 572.628 | **565.952** | 693.561 |
| KITTI driving | PSNR **(higher/better)** | 11.325 | 11.585 | 13.988 | **14.522** | 14.262 | 14.516 |
| | SSIM **(higher/better)** | 0.261 | 0.263 | 0.37 | 0.405 | 0.389 | **0.408** |
| | Cos. Sim. **(higher/better)** | 0.465 | 0.475 | 0.597 | 0.617 | 0.600 | **0.621** |
| | FVD **(lower/better)** | 2921.798 | 2871.245 | 2063.228 | 2127.124 | 2151.003 | **2021.726** |

**[R1, R4] Comparison with SOTA Architectures:** SAVP is a competitive video prediction model that combines many of the previously proposed methods (optical flow, adversarial losses, masks) but it also requires significant hyperparameter tuning. Although SAVP achieved strong results on (relative easy) BAIR Robot Pushing and KTH datasets, it has not been demonstrated on more complex datasets (e.g., BAIR Towel-Pick and Human3.6M are much more challenging than BAIR Robot Pushing and KTH, respectively). In our initial experiments based on the authors' implementation of SAVP, our large-scale models outperformed SAVP. We will further verify this with additional hyperparameter tuning for SAVP and report the results, but as of now, there is no evidence that SAVP (without scaling up) can be competitive to our best performing large-scale models on these challenging datasets. Scaling up SAVP could be interesting future work, but it may be nontrivial due to the complexity of the architecture and hyperparameter tuning.

**[R1] Model capacity comparisons in main text:** Thanks, we will fit the primary capacity results in the main paper.

[Meta-Review · NeurIPS 2019]

All reviewers agree that the paper offers a solid contribution on evaluating the capacity of large neural nets for video prediction tasks. They authors examine a variety of different settings, and provide interesting results. One suggestion for improvement is that the authors evaluate SOTA networks on more complex data sets, and cases where some scenes have higher uncertainty.